# Circa: Stochastic ReLUs for Private Deep Learning

**Zahra Ghodsi**[1], **Nandan Kumar Jha**[2], **Brandon Reagen**[2], **Siddharth Garg**[2]

[1]University of California San Diego, [2]New York University

zghodsi@ucsd.edu, {nj2049, bjr5, sg175}@nyu.edu

## Abstract

The simultaneous rise of machine learning as a service and concerns over user privacy have increasingly motivated the need for private inference (PI). While recent work demonstrates PI is possible using cryptographic primitives, the computational overheads render it impractical. State-of-art deep networks are inadequate in this context because the source of slowdown in PI stems from the ReLU operations whereas optimizations for plaintext inference focus on reducing FLOPs. In this paper we re-think ReLU computations and propose optimizations for PI tailored to properties of neural networks. Specifically, we reformulate ReLU as an approximate sign test and introduce a novel truncation method for the sign test that significantly reduces the cost per ReLU. These optimizations result in a specific type of stochastic ReLU. The key observation is that the stochastic fault behavior is well suited for the fault-tolerant properties of neural network inference. Thus, we provide significant savings without impacting accuracy. We collectively call the optimizations Circa and demonstrate improvements of up to $4.7\times$ storage and $3\times$ runtime over baseline implementations; we further show that Circa can be used on top of recent PI optimizations to obtain $1.8\times$ additional speedup.

## 1 Introduction

Today, Machine Learning as a Service (MLaaS) provides high-quality user experiences but comes at the cost of privacy—clients either share their personal data with the server or the server must disclose its model to the clients. Ideally, both the client and server would preserve the privacy of their inputs and model without sacrificing quality. A recent and growing body of work has focused on designing and optimizing cryptographic protocols for private inference (PI). With PI, MLaaS computations are performed obliviously; without the server seeing the client's data nor the client learning the server's model. PI protocols are built using cryptographic primitives including homomorphic encryption (HE), Secret Sharing (SS), and secure multiparty computation (MPC). The challenge is that all known protocols for PI incur impractically high overheads, rendering them unusable.

Existing PI frameworks [1, 2, 3] are based on *hybrid* protocols, where different cryptographic techniques are used to evaluate different network layers. Delphi [3], a leading solution based on Gazelle [2], uses additive secret sharing for convolution and fully-connected layers. Secret sharing supports fast addition and multiplication by moving large parts of the computation to an offline phase [4]. Thus, convolutions can be computed at near plaintext speed. Non-linear functions, notably ReLU, cannot enjoy the same speedups. Most protocols (including Delphi, Gazelle, and MiniONN [1]) use Yao's Garbled Circuits (GC) [5] to process ReLUs. GCs allow two parties to collaboratively and privately compute arbitrary Boolean functions. At a high-level, GCs represent functions as encrypted two-input truth tables. This means that computing a function with GCs requires the function be decomposed into a circuit of binary gates that processes inputs in a bit-wise fashion. Thus, evaluating ReLUs privately is extremely expensive, to the point that PI inference runtime is dominated by ReLUs [3, 6].

35th Conference on Neural Information Processing Systems (NeurIPS 2021).

Therefore, reducing ReLU cost is critical to realizing practical PI. There are two general approaches for minimizing the cost of ReLUs: designing new architectures that limit ReLU counts, and optimizing the cost per ReLU. Prior work has almost exclusively focused on minimizing ReLU counts. Work along this line includes replacing or approximating ReLUs with quadratics or polynomials (e.g., CryptoNets [7], Delphi [3], SAFENet [8]), designing new networks architectures to maximize accuracy per ReLU (e.g., CryptoNAS [6]), and more aggressive techniques that simply remove ReLUs from the network (e.g., DeepReDuce [9]). Relatively little attention has been given to minimizing the cost of the ReLU operation itself.

In this paper we propose Circa[1], a novel method to reduce ReLU cost based on a new *stochastic ReLU* function. First, we refactor the ReLU as a sign computation followed by a multiplication, allowing us to push the multiplication from GC to SS, leaving only a sign computation in the GC. Next, we approximate the sign computation to further reduce GC cost. This approximation is not free; it results in stochastic sign evaluation where the results are sometimes incorrect (we call incorrect computations faults to differentiate from inference error/accuracy). Finally, we show that stochastic sign can be optimized even further by truncating its inputs; truncation introduces new faults, but only for small positive or negative values.

Our key insight is that deep networks are highly resilient to stochastic ReLU fault behavior, which provides significant opportunity for runtime benefit. The stochastic ReLU introduces two types of faults. First, the sign of a ReLU can be incorrectly computed with probability proportional to the magnitude of the input (this probability is the ratio of the input magnitude over field prime.) In practice we find this rarely occurs as most ReLU inputs are very small (especially compared to the prime) and thus the impact on accuracy is negligible. Second, truncation can cause *either* small positive or small negative values to fault. Circa allows users to choose between these two probabilistic fault modes. In *NegPass*, small negative numbers experience a fault with some probability and are incorrectly passed through the ReLU. Alternatively, *PosZero* incorrectly resolves small positive inputs to zero. Empirically, we find deep networks to be highly resilient against such faults, tolerating more than 10% fault rate without sacrificing accuracy. Compared to Delphi, Circa-optimized networks run up to $3\times$ times faster. We further show that Circa is orthogonal to the current best practice for ReLU count reduction [9]. When combined, we observe an additional $1.8\times$ speedup.

## 2 Background

### 2.1 Private Inference

We consider a client-server model where the client sends their input to the server for inference using the server's model. The client and the server wish to keep both the input and model private during the inference computation. Our threat model is the same as prior work on private inference [1, 2, 3]. More specifically, we operate in a two-party setting (i.e., client and server) where participants are honest-but-curious—they follow the protocol truthfully but may try to infer information about the other party's input/model during execution.

We take the Delphi protocol [3] as a baseline and implement our optimizations over it. Delphi uses HE and SS for linear layers, where computationally expensive HE operations are performed in an offline phase and online computations only require lightweight SS operations. For non-linear activations, Delphi uses ReLUs, evaluated using GCs, and polynomial activations ($x^2$), evaluated using Beaver multiplication triples [4]. While Circa does not use polynomial activations, it uses Beaver triples in its stochastic ReLU implementation. Next, we introduce the necessary cryptographic primitives and provide an overview of the Delphi protocol.

### 2.2 Cryptographic Primitives

**Finite Fields.** The cryptographic primitives described subsequently operate on values in a finite field of integer modulo a prime $p$, $\mathbb{F}_p$, i.e., the set $\{0, 1, \ldots, p-1\}$. In practice, positive values will be represented with integers in range $[0, \frac{p-1}{2})$, and negative values will be integers in range $[\frac{p-1}{2}, p)$.

**Additive Secret Sharing.** Additive secret shares of a value $x$ can be created for two parties by randomly sampling a value $r$ and setting shares as $\langle x \rangle_1 = r$ and $\langle x \rangle_2 = x - r$. The secret can be

---

[1]In Latin, "circa" means approximately.

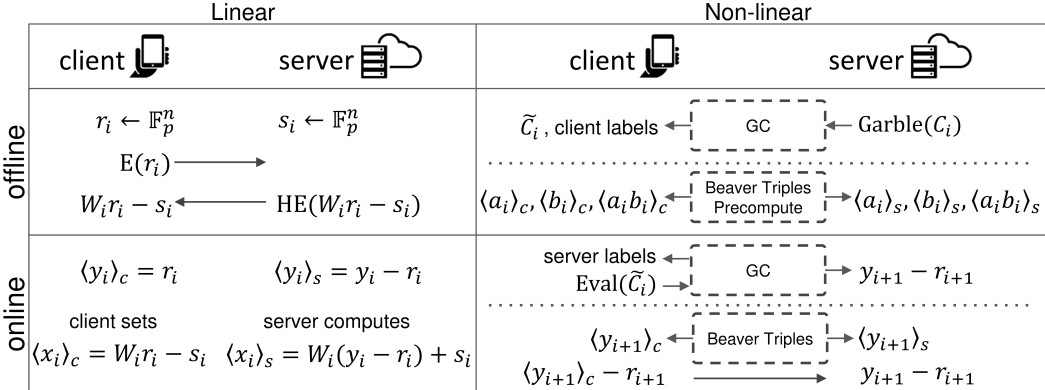

Figure 1: An illustration of the Delphi protocol, which Circa uses. Delphi is a hybrid protocol that uses HE (offline) and SS (online) for linear layers, and GC and Beaver multiplication triples for ReLU and polynomial non-linear layers respectively.

reconstructed by adding the shares $x = \langle x \rangle_1 + \langle x \rangle_2$. Performing additions over two shared values is straightforward in this scheme, each party simply adds their respective shares to obtain an additive sharing of the result.

**Beaver Multiplication Triples.** This protocol [4] is used to perform multiplications over two secret shared values. A set of multiplication triples are generated offline from random values $a$ and $b$, such that the first party receives $\langle a \rangle_1, \langle b \rangle_1, \langle ab \rangle_1$, and the second party receives $\langle a \rangle_2, \langle b \rangle_2, \langle ab \rangle_2$. In the online phase $x$ and $y$ are secret shared among parties such that the first party holds $\langle x \rangle_1, \langle y \rangle_1$ and the second party holds $\langle x \rangle_2, \langle y \rangle_2$. To perform multiplication they consume a set of triples generated offline and at the end of the protocol the first party obtains $\langle xy \rangle_1$ and the second party obtains $\langle xy \rangle_2$.

**Homomorphic Encryption.** HE [10] is a type of encryption that enables computation directly on encrypted data. Assuming a public key $k_{pub}$ and corresponding secret key $k_{sec}$, an encryption function operates on a plaintext message to create a ciphertext $c = E(m, k_{pub})$, and a decryption function obtains the message from the ciphertext $m = D(c, k_{sec})$. An operation $\odot$ is homomorphic if for messages $m_1, m_2$ we have a function $\star$ such that decrypting the ciphertext $E(m_1, k_{pub}) \star E(m_2, k_{pub})$, which we also write as $HE(m_1 \odot m_2)$, gives $m_1 \odot m_2$.

**Garbled Circuits.** GCs [5] enable two parties to collaboratively compute a Boolean function on their private inputs. The function is first represented as a Boolean circuit $C$. One party (the *garbler*) encodes the circuit using procedure $\tilde{C} \leftarrow Garble(C)$ and sends it to the second party (the *evaluator*). The evaluator also obtains labels of the inputs and is able to evaluate the circuit using procedure $Eval(\tilde{C})$ without learning intermediate values. Finally, evaluator will share the output labels with the garbler, and both parties obtain the output in plaintext. The cost of GC is largely dependent on the size of the Boolean circuit being computed.

### 2.3 Delphi Protocol

We now briefly describe the linear and non-linear components of the Delphi protocol, depicted also in Figure 1. For simplicity, we will focus on the $i^{th}$ layer, with input $\mathbf{y_i}$ and output $\mathbf{y_{i+1}}$. The layer performs linear computation $\mathbf{x_i} = \mathbf{W_i}.\mathbf{y_i}$ (here $\mathbf{W_i}$ are the server's weights) followed by a non-linear activation $\mathbf{y_{i+1}} = \text{ReLU}(\mathbf{x_i})$. The protocol consists of an input independent *offline phase*, and an input dependent *online phase*. As a starting point, the client samples random vectors $\mathbf{r_i} \in \mathbb{F}_p^n$ in the offline phase. For input $\mathbf{y_1}$, the client computes $\mathbf{y_1} - \mathbf{r_1}$ and sends it to the server. For the $i^{th}$ layer, the client and the server start with secret shares of the layer input, $\langle \mathbf{y_i} \rangle_c = \mathbf{r_i}$ and $\langle \mathbf{y_i} \rangle_s = \mathbf{y_i} - \mathbf{r_i}$, and use the Delphi protocol to obtain shares of the output, $\langle \mathbf{y_{i+1}} \rangle_c = \mathbf{r_{i+1}}$ and $\langle \mathbf{y_{i+1}} \rangle_s = \mathbf{y_{i+1}} - \mathbf{r_{i+1}}$.

**Linear computation:** In the offline phase, the server samples random vectors $\mathbf{s_i} \in \mathbb{F}_p^n$. Using HE on the server side, as shown in Figure 1, the client then obtains $\mathbf{W_i}.\mathbf{r_i} - \mathbf{s_i}$ without learning the server's randomness or weights and without the server learning the client's randomness. In the *online* phase, the server computes $\mathbf{W_i}.(\mathbf{y_i} - \mathbf{r_i}) + \mathbf{s_i}$, at which point the client and the server hold additive secret shares of $\mathbf{x_i} = \mathbf{W_i}.\mathbf{y_i}$. Circa uses the same protocol for linear layers as Delphi.

**Non-linear computation:** In this description we will focus on ReLU computations (for completeness, Figure 1 also illustrates how quadratic activations are computed). During the offline phase, the server creates a Boolean circuit $C$ for each ReLU in the network, garbles the circuit and sends it to the client along with labels corresponding to the client's input. In the online phase, the linear layer protocol produces the server's share of the ReLUs input. The server sends labels corresponding to its share to the client. The client then evaluates the GC, which outputs the server's share, $\mathbf{y_{i+1}} - \mathbf{r_{i+1}}$, for the next linear layer. Online GC evaluation is the most expensive component of Delphi's online PI latency. Circa focuses on reducing this cost.

## 3 Circa Methodology

We now describe Circa, beginning with a cost analysis of the GC design used in prior work. We then describe three optimizations to reduce GC size and latency that form the core of Circa.

### 3.1 Cost Analysis of ReLU GC

The inputs to a conventional ReLU GC are the client's and server's shares of $x$, i.e., $\langle x \rangle_c$ and $\langle x \rangle_s$, and random value $r$ from the client. Each is a value in the field $\mathbb{F}_p$, implemented using $m = \lceil log(p) \rceil$ bits. Prior work [1, 2, 3] implements ReLU with a circuit that performs several computations contributing to the GC cost as shown in Figure 2(a). First, $x = \langle x \rangle_c + \langle x \rangle_s \bmod p$ is computed by obtaining $z = \langle x \rangle_c + \langle x \rangle_s$ and $z - p$ using two m-bit adder/subtractor modules (ADD/SUB). $z$ is checked for overflow, and either $z$ or $z - p$ is selected using a multiplexer (MUX). Then, $x$ is compared with $\frac{p}{2}$ using an m-bit comparator (>), and a MUX outputs either $0$ or $x$. Finally, the GC outputs the server's share of ReLU$(x)$ by performing a modulo subtraction of $r$ from the output of the previous MUX using two ADD/SUB modules and another MUX.

This design, used by Delphi and Gazelle, results in a GC size of 17.2KB per ReLU. Overall, the GCs for ResNet32, as implemented in Delphi, require close to 5GB of client-side storage per inference[2]. GC size directly correlates with PI latency, resulting in prohibitive online runtime.

### 3.2 Circa's Stochastic ReLU

Circa's Stochastic ReLU is built using three optimizations that work together to reduce GC size. We describe each optimization below.

**Refactoring ReLUs.** Our first observation is that ReLU$(x)$ can be refactored as $x.\text{sign}(x)$, where $\text{sign}(x)$ equals 1 if $x \geq 0$ and 0 otherwise. Since multiplications can be evaluated cheaply online using Beaver triples, only $\text{sign}(x)$ must be implemented in GC. Let $v = \text{sign}(x)$; the GC computes the server's share of $v$, $\langle v \rangle_s = \text{sign}(x) - r$, using shares of $x$ and random value $r$ provided by the client. The client will then set its share to $\langle v \rangle_c = r$.

Figure 2(b) shows our naive GC implementation for $\text{sign}(x)$. As in the ReLU$(x)$ GC (Figure 2(a)), we first compute $x = \langle x \rangle_c + \langle x \rangle_s \bmod p$ using two ADD/SUB modules and a MUX, and use a comparator to check $x$ against $\frac{p}{2}$. By having the client pre-compute and provide both $-r$ and $1 - r$ as inputs to the GC we save two ADD/SUB modules, since we no longer need to perform these computations inside the GC. Note that the client selects $r$ and can compute $-r$ and $1 - r$ by itself at plaintext speed. Formally, our GC for the sign function implements:

$$sign\big(\langle x \rangle_c, \langle x \rangle_s, -r, 1-r\big) = \begin{cases} -r & \text{if } \langle x \rangle_c + \langle x \rangle_s \bmod p > \frac{p}{2} \\ 1-r & \text{otherwise} \end{cases} \tag{1}$$

The client and server now hold secret shares of both $x$ (i.e., $\langle x \rangle_c$ and $\langle x \rangle_s$), and $v$ (i.e., $\langle v \rangle_c$ and $\langle v \rangle_s$). We now use pre-computed Beaver multiplication triples, as described in Section 2.2, to compute shares of $y = x.\text{sign}(x)$. This multiplication is cheap and the optimized ReLU is smaller and faster than standard GC implementation in Figure 2(a), providing modest benefits.

---

[2]Note that a separate GC must be constructed for each ReLU in a network and GCs cannot be reused across inferences.

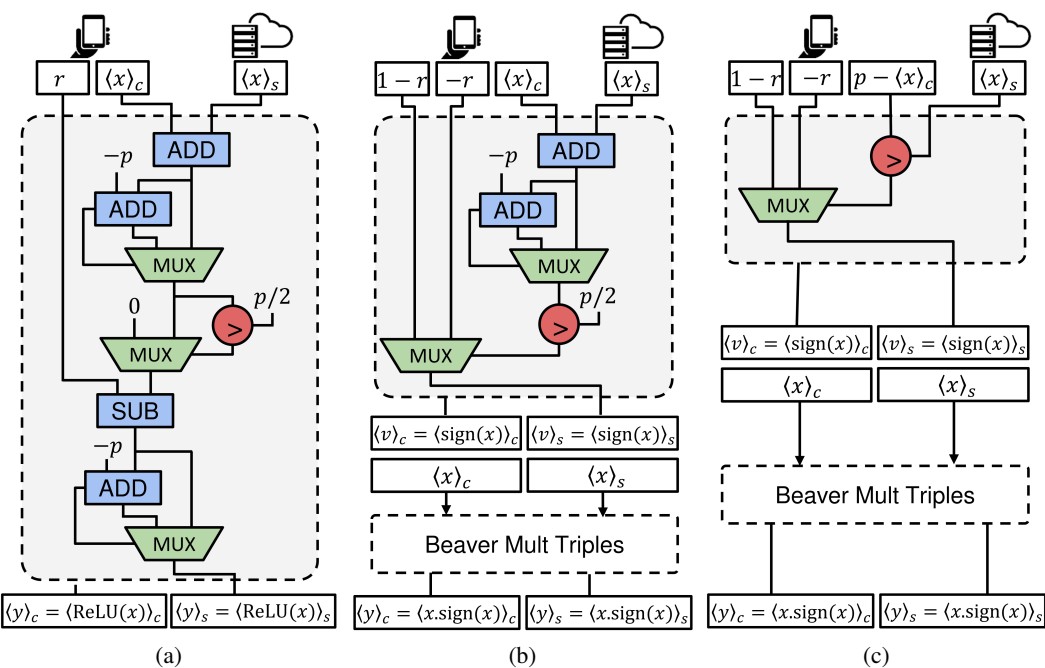

Figure 2: Comparing implementations of the ReLU function for PI. (a) depicts the implementations in prior work [1, 2, 3] where the ReLU function is implemented with GC, (b) shows a naive implementation of the sign function followed by multiplication triples, and (c) describes the implementation in Circa with an optimized sign function followed by multiplication triples. The heaviest part of the computation in each of these implementations relates to the GC which is shown in shaded blocks.

**Stochastic Sign.** Our second optimization further reduces the cost of the sign computation. As noted previously, the naive sign GC still uses high-cost components inside the GC because of the need to perform *modulo* additions to exactly reconstruct $x$; modulo additions require expensive checks for overflow and a subsequent subtraction. Our next optimization only looks at the regime without overflow, greatly simplifying the GC at the cost of introducing occasional faults in sign computation.

Figure 2(c) shows our proposed stochastic sign optimization that reduces the logic inside the GC to only a comparator and a MUX. We first formally define the stochastic sign function.

$$\widetilde{sign}\big(p - \langle x \rangle_c, \langle x \rangle_s, -r, 1 - r\big) = \begin{cases} -r & \text{if } \langle x \rangle_s \leq p - \langle x \rangle_c \\ 1 - r & \text{otherwise} \end{cases} \tag{2}$$

Note that in the stochastic sign GC, the client sends the negated value of its share (or $p - \langle x \rangle_c$) instead of the share directly. This optimization avoids the need to compute $p - \langle x \rangle_c$ inside the GC itself. We formalize the fault rates of the stochastic ReLU below.

**Theorem 3.1.** *For any $x \in \mathbb{F}_p$, assuming shares $\langle x \rangle_s = x + t \bmod p$ and $\langle x \rangle_c = p - t$ where $t$ is picked uniformly at random from $\mathbb{F}_p$,*

$$P\Big\{ \widetilde{sign}\big(\langle x \rangle_s, p - \langle x \rangle_c, -r, 1 - r\big) \neq sign\big(\langle x \rangle_c, \langle x \rangle_s, -r, 1 - r\big) \Big\} = \frac{|x|}{p}.$$

*Proof.* Consider the case where $x$ is positive, i.e., $x \leq \frac{p}{2}$. The wrong sign is assigned to $x$ if $\langle x \rangle_s \leq p - \langle x \rangle_c$, which can be rewritten as $x + t \bmod p \leq t$. This is true when adding $x$ and $t$ incurs an overflow, with $t \geq p - x$. Since $t$ is drawn at random, the probability of error $P = \frac{x}{p}$. A similar analysis for negative values of $x$ shows that a wrong sign is assigned when $x + t < p$ which results an error probability of $P = \frac{|x|}{p}$, where $|x| = p - x$ for $x > \frac{p}{2}$. $\qquad \square$

**Truncated Stochastic Sign.** Our third and most effective optimization builds on the observation that the $\langle x \rangle_s \leq p - \langle x \rangle_c$ (equivalently, $\langle x \rangle_s \leq t$) check in the stochastic sign GC can be performed on

truncated values. We show that truncation introduces an additional fault mode; in particular, the check is incorrect with some probability for small positive values of $x$ in the range $[0, 2^k)$ (i.e., values that truncate to 0 with $k$-bit truncation) but is correct for all other values of $x$.

Let $\lfloor x \rfloor_k$ represent truncation of the $k$ least significant bits of $x$, i.e., only the $m-k$ most significant bits of $x$ are retained. We define a truncated stochastic sign as:

$$\widetilde{sign}_k\big(p - \langle x \rangle_c, \langle x \rangle_s, -r, 1-r\big) = \widetilde{sign}\big(\lfloor p - \langle x \rangle_c \rfloor_k, \lfloor \langle x \rangle_s \rfloor_k, -r, 1-r\big) \tag{3}$$

where $k$ represents the amount of truncation. We now prove that the truncated stochastic sign function incurs additional errors (over the stochastic sign) only for small positive values.

**Theorem 3.2.** *For any* $x \in \mathbb{F}_p$, *assuming shares* $\langle x \rangle_s = x + t \bmod p$ *and* $\langle x \rangle_c = p - t$ *where $t$ is picked uniformly at random from* $\mathbb{F}_p$, *and assuming* $\widetilde{sign}\big(\langle x \rangle_c, p - \langle x \rangle_s, -r, 1-r\big) = sign\big(\langle x \rangle_c, \langle x \rangle_s, -r, 1-r\big)$, *then:*

$$P\Big\{\widetilde{sign}_k\big(\langle x \rangle_c, p - \langle x \rangle_s, -r, 1-r\big) \neq \widetilde{sign}\big(\langle x \rangle_c, p - \langle x \rangle_s, -r, 1-r\big)\Big\} = \frac{2^k - |x|}{2^k} \quad \forall x \in [0, 2^k),$$

*and zero otherwise.*

*Proof.* For negative $x$, the stochastic sign is error-free if $\langle x \rangle_s \leq p - \langle x \rangle_c$ (equivalently $\langle x \rangle_s \leq t$). Consequently the truncated stochastic sign would not incur an error, since $\lfloor \langle x \rangle_s \rfloor_k = \lfloor t \rfloor_k$ is assigned a negative sign. The additional error in this case is when $\lfloor \langle x \rangle_s \rfloor_k = \lfloor t \rfloor_k$ for positive values of $x$. Therefore the error happens when $\lfloor x + t \rfloor_k = \lfloor t \rfloor_k$, or $x + t$ does not overflow to higher $p - k$ bits. So for $|x| > 2^k$ there is no error. In other case, the error happens when $x + t > 2^k$ or equivalently $t > 2^k - |x|$. Assuming a uniform distribution the error probability would be $P = \frac{2^k - |x|}{2^k}$. $\square$

**Putting it All Together: the Stochastic ReLU.** We define Circa's stochastic ReLU as $\widetilde{ReLU}_k(x) = x.\widetilde{sign}_k(x)$ with $\widetilde{sign}_k$ defined in Eq. 3. Stochastic ReLUs incur *two* types of faults: (1) a sign error, independent of $k$, with probability $\frac{|x|}{p}$ (in practice $|x| \ll p$ for typical choices of prime), and (2) *small* positive values in the truncation range $x \in [0, 2^k)$ are zeroed out with high probability $\frac{2^k - |x|}{2^k}$; however, for small values we expect the impact on network accuracy to be low.

We note that Eq. 2 could have been defined such that $\widetilde{sign}$ outputs $-r$ for $\langle x \rangle_s < p - \langle x \rangle_c$. With this modification, truncation errors occur with the same probability but for small negative values in range $[p - 2^k, p)$ that are passed through to the ReLU output. That is, our stochastic ReLU can operate in one of two modes: (1) **PosZero**, that zeros out small positive values, or (2) **NegPass**, that passes through small negative values.

## 4 Evaluation

In this section we evaluate Circa and validate our error model. We show that our optimizations have minimal effect on network accuracy and show the runtime and storage benefits of Circa.

### 4.1 Experimental Setup

We perform experiments on ResNet18 [11], ResNet32 [11] and VGG16 [12]. Since Circa can be used to replace ReLU activations in *any* network, we also perform experiments on DeepReDuce-optimized models [9] that are the current state of the art for fast PI. We train these networks on CIFAR-10/100 [13] and TinyImageNet [14] datasets in plaintext (Circa is not involved). The training procedure uses stochastic gradient descent with step learning optimizer and 0.1 initial learning rate, 128 batch size, 0.0001 weight deacy, 0.9 momentum, and milestones at 100th and 150th epochs for 200 epochs. CIFAR-10/100 (C10/100) datatset has 50k training and 10k test images (size $32 \times 32$) separated into 10/100 output classes. TinyImageNet (Tiny) consist of 200 output classes with 500 training and 50 test samples (size $64 \times 64$) per class. Prior work on PI typically evaluate on smaller datasets and lower resolution images because PI is prohibitively expensive for ImageNet or higher resolution inputs.

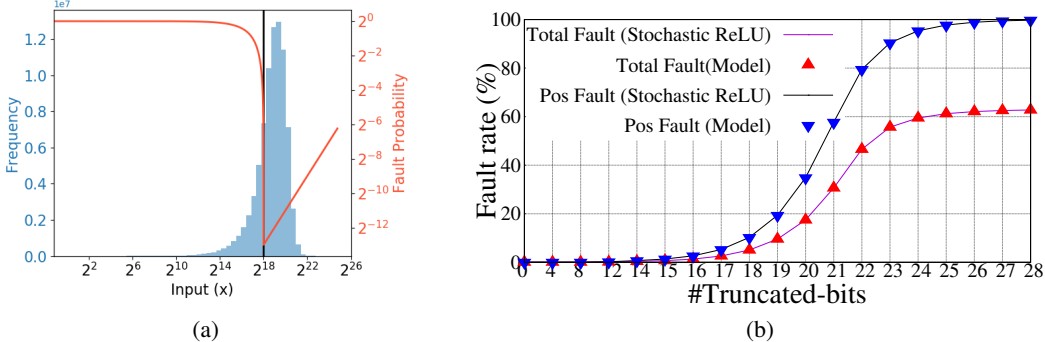

(a)                                                                                          (b)

Figure 3: (a) Fault probability of stochastic ReLU with 18-bit truncation in the PosZero mode, and a histogram of ResNet18's activations, (b) Validating Circa's fault model for ResNet18 on CIFAR-100.

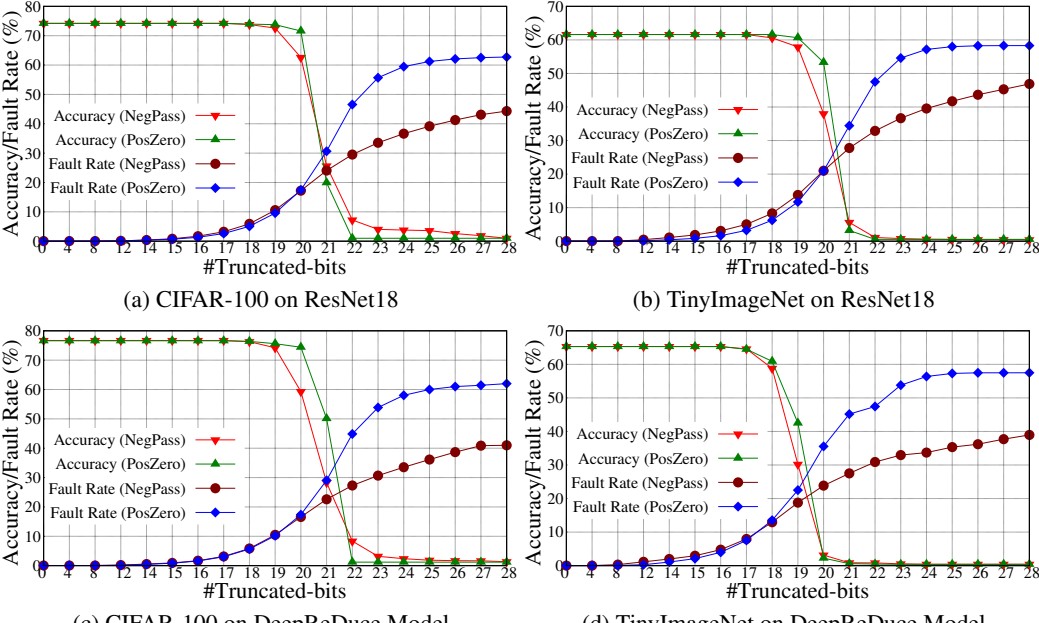

(a) CIFAR-100 on ResNet18                    (b) TinyImageNet on ResNet18

(c) CIFAR-100 on DeepReDuce Model          (d) TinyImageNet on DeepReDuce Model

Figure 4: Accuracy and fault rate variation with the increasing number of truncated-bits. Experiments performed on CIFAR-100 and TinyImageNet with ResNet18 vanilla model (top row) and with DeepReDuce models [9] (bottom row).

Circa uses the Delphi protocol as a base, but substantially modifies the way ReLUs are implemented. We use the SEAL library [15] for HE, and fancy-garbling library [16] for GC. We benchmark PI runtime on an Intel i9-10900X CPU running at 3.70GHz with 64GB of memory. The baseline accuracy of the models in PI is reported using an integer model with network values in a prime field. To obtain an integer model, we scale and quantize model parameters and input to 15 bits (as in Delphi), and pick a 31 bit prime field ($p = 2138816513$) to ensure that multiplication of two 15-bit values does not exceed the field. The baseline accuracy of our integer models is reported in Table 1.

### 4.2 Experimental Results

**Validating the Stochastic ReLU Fault Model.** We begin by validating our model of stochastic ReLUs described in Section 3.2 (Theorem 3.1 and 3.2) against Circa's stochastic ReLU implementation. Figure 3(a) plots the fault probability of stochastic ReLU with 18-bit truncation in the PosZero mode against the histogram of ResNet18's activations after the first convolution layer. According to our fault model, small positive activations in truncation range ($0 \leq x < 2^{18}$) incur a high fault probability, and we have $P = (2^{18} - x)/2^{18}$. For values outside of this range, the fault probability is small and grows proportional to activation absolute value, and we have $P = |x|/p$. Figure 3(b) plots fault rates on ResNet18 trained on C100 for PosZero sotchastic ReLU mode. We plot the total fault rate for all

Table 1: Circa on CIFAR-10 (C10), CIFAR-100 (C100), and TinyImageNet (Tiny).

| Network-Dataset | #ReLUs (K) | Baseline Acc | Stochastic ReLU | | Baseline Runtime (s) | Circa Runtime (s) | Runtime Speedup |
| --- | --- | --- | --- | --- | --- | --- | --- |
| | | | NegPass (bits) | PosZero (bits) | | | |
| ResNet32-C10 | 303.1 | 92.43% | 91.47% (12) | 91.85% (12) | 6.32 | 2.47 | 2.6× |
| ResNet18-C10 | 557.1 | 94.66% | 93.77% (11) | 94.24% (11) | 11.05 | 3.89 | 2.8× |
| VGG16-C10 | 284.7 | 94.00% | 93.77% (12) | 93.61% (13) | 5.89 | 2.25 | 2.6× |
| ResNet32-C100 | 303.1 | 67.32% | 66.41% (14) | 66.32% (13) | 6.32 | 2.47 | 2.6× |
| ResNet18-C100 | 557.1 | 74.24% | 73.80% (13) | 73.76% (12) | 11.05 | 4.15 | 2.7× |
| VGG16-C100 | 284.7 | 73.94% | 73.25% (12) | 73.19% (12) | 5.89 | 2.25 | 2.6× |
| ResNet32-Tiny | 1212.4 | 55.53% | 55.15% (16) | 54.56% (15) | 24.24 | 9.04 | 2.7× |
| ResNet18-Tiny | 2228.2 | 61.60% | 60.60% (13) | 60.65% (12) | 44.55 | 14.28 | 3.1× |
| VGG16-Tiny | 1114.1 | 50.85% | 50.73% (12) | 50.30% (12) | 21.41 | 6.96 | 3.1× |

activations and the fault rate for only positive activations. Points in the plot indicate measurements from the implementation and the lines show our estimates using the model. We observe that our model is consistent with the implementation across a wide range of truncation values. As expected, as we increase the amount of truncation, the fault rates increase. With 28 bits of truncation, all positive activations are faulty. The total fault rate is 60%, which is lower than the positive fault rate because negative activations incur relatively few faults, as predicted by our model.

**Fault Rates and Test Error vs. Truncation.** Figure 4 shows the relationship between truncation, fault rates, and test accuracy. The experiments are done using the C100 and Tiny datsets with ResNet18 and DeepReDuce-optimized networks. Each plot shows data for both NegPass and PosZero modes. We observe that in all cases, Circa is able to truncate 17-19 bits with negligible accuracy loss at fault rates up to 10%. We also find that the PosZero version of Circa is consistently slightly better than NegPass for these models, enabling between 1-2 extra bits of truncation. For all subsequent experiments, we pick the fault mode and truncation such that the resulting accuracy is within 1% of baseline.

**Circa Accuracy and PI Runtime on Baseline Models.** Table 1 shows the accuracy and runtime of Circa for the C10/100 and Tiny datasets applied on top of standard ResNet18/32 and VGG16 models. Circa achieves 2.6× to 3.1× PI runtime improvement, in each instance with less than 1% accuracy reduction. The runtime improvements are larger for Tiny because the baseline networks are different owing to Tiny's higher spatial resolution.

**Circa vs. State of the Art.** Circa's baseline accuracy is obtained from plaintext training and can be different from other frameworks. For this reason, we compare relative accuracy drops from baseline using a 1% accuracy drop as a target. We also provide comparisons at similar baseline accuracy to other frameworks. Delphi, the baseline protocol on which we build Circa, reduces PI runtime by replacing selected ReLUs with cheaper quadratic activations. For C100, Delphi reduces the number of ReLUs in ResNet32 by 1.2× with 1% accuracy loss compared to baseline which, at best, translates to an equal reduction in PI latency. For the same setting, Circa achieves a 2.6× reduction in PI latency. SAFENet [8] achieves 1.9× speedup over Delphi on ResNet32/C100, while Circa achieves a 2.6× speedup. To fairly compare absolute accuracies, we also trained a ResNet32 model on C100 using cosine annealing learning rate for 90 epochs with a 68.15% baseline accuracy that is closer to the baseline in Delphi and SAFENet. Using this model, Circa achieves 67.76% accuracy (a 0.39% accuracy drop) for 2.6× speedup (17-bit truncation), while SAFENet and Delphi have 67.5% and 67.3% accuracies for 1.9× and 1.2× speedups, respectively.

Circa can be applied on *any* pre-trained ReLU network. Table 2 shows Circa's accuracy and PI runtime applied to DeepReDuce-optimized networks, the current state of the art in PI, across a range of ReLU counts on C100 and Tiny. Circa reduces DeepReDuce PI latency by 1.6× to 1.8×, with less than 1% accuracy loss. Moreover, Circa improves the set of Pareto optimal points. For example, Circa achieves 75.34% accuracy on C100 with 1.65s PI runtime, while DeepReduce has both higher runtime (1.71s) and lower accuracy (74.7%). For Tiny, Circa increases DeepReduce's accuracy from 59.18% to 61.63% at effectively iso-runtime (3.18s vs. 3.21s).

Finally, NASS [17] builds on top of Gazelle and uses a reinforcement learning agent to search for optimal architectures and quantization parameters for each layer. The quantization approach in NASS primarily reduces the linear layer costs, which are already low for Circa (heavy HE operations for linear layers are moved offline per Delphi's protocol). On the other hand, the

Table 2: Circa with DeepReDuce (ResNet18) models on CIFAR-100 and TinyImageNet.

| Network-Dataset | #ReLUs (K) | Baseline Acc | Stochastic ReLU | | Baseline Runtime (s) | Circa Runtime (s) | Runtime Speedup |
| --- | --- | --- | --- | --- | --- | --- | --- |
| | | | NegPass (bits) | PosZero (bits) | | | |
| DeepReD1-C100 | 229.4 | 76.22% | 76.34% (13) | 75.62% (12) | 3.18 | 1.84 | 1.7× |
| DeepReD2-C100 | 114.7 | 74.72% | 73.47% (13) | 73.64% (13) | 1.71 | 1.05 | 1.6× |
| DeepReD3-C100 | 196.6 | 75.51% | 75.13% (13) | 75.34% (13) | 2.76 | 1.65 | 1.7× |
| DeepReD4-C100 | 98.3 | 71.95% | 71.45% (13) | 71.65% (13) | 1.48 | 0.903 | 1.6× |
| DeepReD1-Tiny | 917.5 | 64.66% | 64.62% (14) | 64.53% (14) | 12.27 | 6.68 | 1.8× |
| DeepReD2-Tiny | 458.8 | 62.26% | 61.28% (15) | 61.26% (15) | 6.50 | 3.94 | 1.6× |
| DeepReD5-Tiny | 393.2 | 61.65% | 61.63% (15) | 61.66% (15) | 5.38 | 3.21 | 1.7× |
| DeepReD6-Tiny | 229.4 | 59.18% | 58.65% (15) | 58.61% (15) | 3.18 | 2.01 | 1.6× |

GC evaluation cost reduction of NASS is proportional to the reduction in number of ReLUs and is a result of fewer ReLUs in the proposed architecture. Therefore, Circa and NASS can be used in conjunction to further reduce costs in each framework. To demonstrate this, we trained the architecture from NASS and obtained a baseline $86.72\%$ accuracy with 15-bit quantization. Applying Circa over this baseline yields $85.46\%$ accuracy with a $2.2\times$ speedup.

Reducing the quantization level to 10 bits would allow a smaller 20-bit prime (further reducing GC costs). Applying Circa on the 10-bit network yields $84.64\%$ accuracy (compared to $84.6\%$ reported in NASS) at $2.35\times$ speedup over the 15-bit network.

**Effectiveness of Circa Optimizations.** Circa encompasses three optimizations that build on top of each other, buying us multiplicative savings in GC size and PI runtime. Figure 5 shows the GC size after each optimization. Replacing the baseline 31-bit ReLU GC with a 31-bit sign GC reduces GC size by $1.4\times$, (with no accuracy loss), a 31-bit stochastic sign GC is $1.9\times$ smaller, and truncating the stochastic sign to 12-bits achieves $4.7\times$ saving over the baseline. The runtime improvements from each of these optimizations are shown in Table 3 in the Appendix.

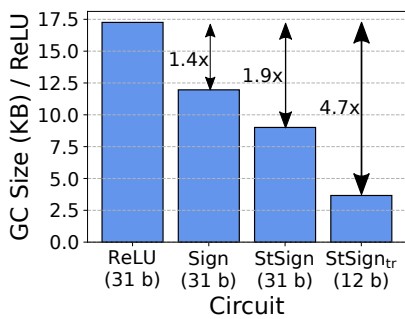

Figure 5: Garbled circuit size comparison between baseline ReLU, naive sign and Circa stochastic ReLUs.

## 5 Related Work

**PI Protocols.** Over the past few years a series of papers have proposed and optimized protocols for private machine learning. CryptoNets [7] demonstrates an HE only protocol for inference using the MNIST dataset. SecureML [18] shows how secret sharing could be used for MNIST inference and trains linear regression models [18]. MiniONN [1] combines secret sharing with multiplication triples and GCs, allowing them to run deeper networks, and forms the foundation for a series of follow-on protocols. While MiniONN generates multiplication triples for each multiplication in a linear layer, Gazelle [2] uses an efficent additive HE protocol to speed up linear layers. Delphi shows how significant speedup can be obtained over Gazelle by moving heavy cryptographic computations offline. XONN [19] enables private inference using only GCs for binarized neural networks and leverages the fact that XORs can be computed for free in the GC protocol to achieve speedups. Another approach is to replace GCs with secure enclaves and process linear layers on GPUs for more performance [20]. Some have also focused on privacy enhanced training [21, 22], typically assuming a different threat model than this work.

**ReLU optimization.** Prior work has also looked at designing ReLU optimized networks. A common approach is to replace ReLUs with quadratics [7, 3, 8, 23]. While effective in reducing GC cost, this complicates the training process and can degrade accuracy. Another approach is to design novel ReLU-optimized architectures. CryptoNAS [6] develops the idea of a ReLU budget and designs new architectures to maximize network accuracy per ReLU. DeepReDuce is recent work that proposes simply removing ReLUs from the network altogether [9]. DeepReDuce is the current state-of-the-art solution for PI, and we demonstrated how Circa can be used on top of it for even more savings.

**Fault tolerant inference.** Many have previously shown neural inference is resilient to faults even during inference. In the systems community, fault tolerant properties are often used to improve

energy-efficiency and runtimes [24, 25, 26, 27]. Others have shown that networks can tolerate approximation to reduce model size by pruning insignificant weights and activations and possibly compressing them [28, 29, 30, 31].

## 6 Conclusion

This paper presented Circa, a new method to significantly speed up PI by lowering the high cost of ReLUs via approximation. Our overarching objective was to minimize the amount of logic that must occur inside the expensive GC. To achieve this goal we reformulated ReLU as an explicit sign test and mask, where only the sign test is evaluated with GCs and showed that we can truncate, or simply remove, many of the least significant bits to the sign test for even more savings. Though the sign test and truncation optimizations introduce error, we rigorously evaluated the effects and found a negligible impact on accuracy. Compared to a baseline protocol (Delphi) for PI, Circa provided over $3\times$ ($2.2\times$ for quadratic Delphi) speedup. Furthermore, we showed how existing state-of-the-art PI optimizations can be combined with Circa for even more savings, resulting in an additional speedup of $1.8\times$ over a baseline protocol. In absolute terms, Circa can run TinyImageNet inferences within 2 seconds, bringing PI another step closer to real-time deployment.

**Limitations and Societal Impact** Circa applies only for inference and not for training, and additionally only applies to certain types of deep networks. Private inference seeks to protect individuals from having to reveal sensitive data, but might also allow unregulated misuse of deep learning services.

## Acknowledgements

This work was supported in part by the Applications Driving Architectures (ADA) Research Center, a JUMP Center co-sponsored by Semiconductor Research Consortium (SRC) and the Defense Advanced Research Projects Agency (DARPA), by the DARPA Data Protection in Virtual Environments (DPRIVE) program (contract HR0011-21-9-0003), and by the National Science Foundation (NSF) grants 1646671 and 1801495. The views, opinions and/or findings expressed are those of the author and should not be interpreted as representing the official views or policies of the Department of Defense or the U.S. Government.

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
