# OpenReview forum: "Circa: Stochastic ReLUs for Private Deep Learning"
_NeurIPS.cc/2021/Conference — NeurIPS 2021 Poster_

### Official Review · Reviewer_9jtr · 2021-07-13

**Rating:** 7
**Confidence:** 2

**Summary:**

The authors propose to improve the speed of modern Homomorphic encryption (HE) approaches which leverage GC for RELUs by moving as much computations as possible outside of the GC. The optimizations introduced in Circa (i.e truncation and removing the modulo addition) come at the cost of stochastic faults. This study demonstrates that modern deep networks are highly resilient to RELU fault behavior and that Circa (the proposed) approach can offer up to 3x faster predictions than Delphi, a state-of-the-art framework.

**Limitations And Societal Impact:**

The authors adequately note their paper's limitations (applicable for inference only and only applicable to RELU networks) and the possible negative social impact.

**Main Review:**

Originality:
This study offers a novel and interesting direction for improving the speed of HE approaches by allowing stochastic faults. It observes that deep networks are resilient to stochastic RELU faults, and this property enables strong speed optimizations without significantly compromising accuracy.

Quality / Significance:
The core claim of this paper, Circa can offer a significant speed boost without compromising accuracy, is well supported by the empirical results. This paper tackles a central and common challenge for HE, i.e estimating relus, and makes a meaningful step towards real time applications of HE.

Clarity:
This paper is clearly written.


**Time Spent Reviewing:**

2

---

> ### Author Response · Authors · 2021-08-11
> **Response**
>
> We thank our reviewer for their time and effort to review our manuscript, and appreciate their positive feedback!

---

### Official Review · Reviewer_DEjr · 2021-07-16

**Rating:** 7
**Confidence:** 4

**Summary:**

This paper proposes Circa to reduce the communicational overhead of ReLU-based privacy-preserving machine learning using three modifications. First, the ReLU circuit is refactorized into two parts including multiplication and sign function, so that a GC-based ReLU is converted into a GC-based Sign function with a Beaver’s Triple’s based multiplication. Secondly, GC-based modular operation is removed from GC-based circuits at the cost of precision. Thirdly, quantization or truncation is used to further speedup GC-based circuits at the cost of precision (potential decryption errors). The authors show that inference accuracy won't be decreased a lot by this decreased precision due to the network’s error-tolerance ability.

**Limitations And Societal Impact:**

-

**Main Review:**

This paper provides a clear ablation study for each of three techniques including sign, stochastic sign, and truncation. For example, readers are easy to find a take-home message that truncation on GC-based circuits plays a very important role in the last performance.

However, there are other comments for the paper.

1. The mechanism of the stochastic sign function is not clear. I knew that this paper tries to replace equation 1 with equation 2 since equation 2 does not have the operation of mod $p$. But it is confusing for the condition in equation 2: if $<x>_s$ <= $p$ - $<x>_c$. This condition may have a conflict with the condition in equation 1. From my understanding, $<x>_c$ +$<x>_s$ =$x$ whose range is from 0 to $2p$.    In equation 1, if $x$ mod $p$ > $p/2$, the output is $-r$, otherwise the output is $1- r$. Since $x$ is from 0 to $2p$, the output is $-r$ if $ x$ $\in$ (0, p/2) $\cup$ (p, 3p/2).  However, in equation 2, if $x$  $\in$ (0, p), the output is $-r$ which is different from equation 1. It is not clear how the proposed work handles this conflict without a large error. The proof of the error probability is not clear.

2. Unconvincing claims and assumptions. In line 198, this paper claims “in practice, $|x| << p$  for typical choice of prime”. However, if the input $x$ is 15 bits, the weights are also 15 bits, and $p$ is 32 bits, the intermediate results $x$ may be near to the $p$, especially when a convolutional layer, normalization layer, or the average pooling layer are used together. It is not clear how the paper controls the $|x|$ to be far less than p. Also, the error probability of stochastic ReLU shows that $p$ should be larger to maintain the small error probability, therefore introducing an overhead about HE parameters selection. For example, schemes without stochastic ReLU could use smaller p, but once using the proposed stochastic ReLU, one must use larger p which potentially introduces larger ciphertext and latency. This paper does not discuss this overhead.

3. Limited function and usage. The proposed work only supports the ReLU function. It is not clear how to use the proposed methods on the other non-linear operations like max-pool, sigmoid, and softmax function which are also very expensive based on GC. It is much more important to support these non-linear operations other than ReLU functions.

4. Prior work. It is good to compare the truncation method with related work NASS [1] which also uses quantization on the network.

I would like to see the paper's presentation in NeurIPS and will consider modifying my score if the authors clarify my concerns well.

[1] Bian, S., Jiang, W., Lu, Q., Shi, Y., & Sato, T. (2020). NASS: Optimizing Secure Inference via Neural Architecture Search. ECAI.


**Time Spent Reviewing:**

8

---

> ### Author Response · Authors · 2021-08-11
> **Response**
>
> We thank our reviewer for their thorough feedback and diligent questions, and will address their comments below.
>
> **Q1:** We show that our error probability proofs follow from the reviewer’s analysis, except for a minor correction in the reviewer’s comment which we clarify later on. For consistency, we use the paper notation by referring to $x$ as the reconstructed value from the shares, i.e., $x = (xc+xs) \bmod{p} \in [0,p)$, and $x’$ as the summation of shares without modulo operation, i.e., $x’=xc+xs \in [0,2p)$. That is, as in the paper, we refer to the $x$ in the reviewer’s question as $x’$, and $x = x’ \bmod{p}$.
>
> We make a small correction to the reviewer’s comment that for negative values of $x$ (i.e., $x > p/2$ and output is $-r$) we have $x’ \in (p/2,p)\cup(3p/2, 2p)$ [not $x’ ∈ (0,p/2)\cup(p, 3p/2)$ as in the comment]. Now indeed, in Equation 2, instead we check if $x’ \in [0,p)$ and output $-r$ instead of the check in Equation 1. In doing so, we can make two possible mistakes:
> 1. if $x$ is positive, i.e.,  $x \in [0, p/2)$ _and_ $x’ \in [0, p/2)$,
> 2. if $x$ is negative,  i.e.,  $x\in [p/2, p)$, _and_ $x’\in(3p/2, 2p)$.
>
> It turns out, as shown in the detailed analysis below, the probability of making these mistakes is $|x|/p$ for any $x$.
>
> Consider Case 1. For Case 1 to happen, $xs’ = x+t$ (note: $xs = xs’ \bmod{p}$) must be greater than $p$. This would imply that $xs = (xs’-p)$ and therefore $x’ = xs+xc = (x+t-p) + (p-t) = x$. If that were not the case, $xs=xs’$ and $x’=x+t+p-t=x+p$, which would not satisfy Case 1.
>
> This means we need $x+t ≥ p$ or equivalently, $t ≥ p-x$. Now, since $t$ is picked uniformly at random from $[0,p)$, therefore $P \\{ t ≥ p-x \\} = x/p$ (note that $x$ is positive). An analogous argument applies for Case 2. We will clarify the proof further in the paper.
>
>
> **Q2:** We note that the choice of a  31-bit prime is identical to prior work [1][8]; for example Delphi [1] also uses 15-bit weights and activations and a 31 bit prime. To avoid overflows, Delphi scales down the result of a multiplication back down to 15-bits before performing another multiplication; therefore, convolution, normalization and pooling layers can be concatenated without overflows. Circa uses exactly the same approach. _That is, Circa does not choose a larger prime to keep error probabilities low, it uses the same prime as the baseline and other protocols_.
>
> Circa also does not explicitly control $|x|$ to keep error probabilities low: how then are we able to operate at low error probabilities? The answer is because as a number of prior works have shown [9], the distribution of activations of state-of-art neural networks is mostly concentrated around small values. Thus, although in theory it is possible to see values of $|x|$ close to $p$, the likelihood of this happening is small. This is demonstrated in the paper in Fig 3(a), which shows the histogram of $x$ for one of the layers in a state-of-art network, and a similar trend is observed for all layers and other networks. We will update the paper clarifying these details and also add more histograms of $|x|$ across neural network layers to empirically demonstrate our claims.
>
> **Q3:** Circa targets ReLU cost reductions because prior work (Delphi [1], SafeNet [2], CryptoNAS [3], DeepReDuce [5]) show that ReLUs dominate PI latency, especially for protocols that use secret sharing for linear layers as Delphi, CryptoNAS, DeepReduce and Circa do. Pooling layers are less sensitive, and average-pooling has been able to replace max-pooling layers in prior work successfully (CryptoNAS [3], HCNN [4]). Additionally, for CNNs, softmax layers are only required at the output (and play their role mostly during training). Therefore, as in prior work, ReLUs are a natural target for optimizing PI inference latency of state-of-the-art CNNs, and we find that optimizing ReLUs alone gives up to 3x speed-up.
>
> Nonetheless, the sign function optimizations in Circa can potentially be generalized to optimize any piecewise linear (PWL) function, or any non-linear function approximated as a PWL function. For example, Circa’s optimizations may be useful in reducing the cost of PWL approximations of sigmoid functions [8][10].
>
> **Q4:** We thank our reviewer for pointing us to this relevant work. We note that Circa and NASS build on different protocols; NASS builds on top of Gazelle protocol, while Circa builds on top of Delphi. A key difference between Delphi and Gazelle is that Delphi significantly reduces the online costs of linear/conv layers, making them much cheaper than Gazelle’s online linear/conv layer costs (see Figure 13b of Delphi).
>
> From our understanding of NASS, its quantization approach primarily reduces the linear/conv layer costs, which are already very low for Circa (since it builds on Delphi). On the other hand, the ReLU/GC cost reductions of NASS come almost entirely from reducing the number of ReLUs in the network; for example, in Table 2 of the NASS paper we can see that both the # of ReLUs and GC runtime go down by ~2x compared to the baseline Gazelle paper. Therefore, as best as we can tell, in NASS, quantization is being used only to reduce linear layer costs, not ReLU costs (and, as stated above, online linear layer costs are already negligible for Delphi/Circa).
>
> Nonetheless, Circa can be used on top of NASS to further reduce its GC costs and vice-versa. We trained the NASS architecture (Table 2 of the NASS paper) and obtained a baseline 86.72% accuracy with 15-bit quantization. Adjusting for any protocol, implementation and hardware platform differences, the GC run-times of this implementation should be comparable to that of the NASS network in Table 2 of the NASS paper (recall again that NASS’ GC runtimes appear to be proportional to ReLU counts and not the extent of quantization).
>
> Applying Circa on top of this baseline yields 85.46% accuracy with a 2.2x speed-up. Further reducing the quantization level to 10-bits allows us to use a smaller 20-bit prime (further reducing GC costs) and applying Circa on the 10-bit quantized network yields a final 84.64% accuracy at 2.35x speed-up compared to the baseline 15-bit network. Therefore the quantization idea from NASS can help reduce Circa’s GC runtimes at the expense of some accuracy drop. Conversely, the Circa speed-ups reported above will directly reduce the GC costs of the NASS network reported in Table 2 while providing comparable accuracy (84.64% vs. 84.% reported in the NASS paper). We will update the paper with this discussion.
>
>
>
>
>
> [1] Mishra et al, “DELPHI: A Cryptographic Inference Service for Neural Networks”, USENIX Security 2020.
>
> [2] Lou et al, “SafeNet: A Secure, Accurate and Fast Neural Network Inference”, ICLR 2021.
>
> [3] Ghodsi et al, “CryptoNAS: Private Inference on a ReLU Budget”, NeurIPS 2020.
>
> [4] Al Badawi et al, “Towards the AlexNet Moment for Homomorphic Encryption: HCNN, the First Homomorphic CNN on Encrypted Data with GPUs”, TETC 2020.
>
> [5] Kumar Jha et al, “DeepReDuce: ReLU Reduction for Fast Private Inference”, ICML 2021.
>
> [6] Mohassel et al, “SecureML: A system for scalable privacy-preserving machine learning”, SP 2017.
>
> [7] Juvekar et al, “Gazelle: A low latency framework for secure neural network inference”, USENIX Security 2018.
>
> [8] Liu et al, “Oblivious Neural Network Predictions via MiniONN Transformations”, CCS 2017.
>
> [9] Cao et al, “Randomly Translational Activation Inspired by the Input Distributions of ReLU”, Neurocomputing 2018.
>
> [10] Boura et al, “High-Precision Privacy-Preserving Real-Valued Function Evaluation”, FC 2018.

---

> > ### Author Response · Authors · 2021-08-31
> > **Available for Discussion**
> >
> > We hope our responses have addressed the reviewer's concerns, but if not we are available/open to addressing any outstanding issues.

---

> > > ### Comment · Reviewer_DEjr · 2021-08-31
> > > **Q1 and Q2**
> > >
> > > Thanks for the authors' responses. Q3 and Q4 are very clear to me. But I still have concerns about Q1 and Q2.
> > >
> > > For Q1, thanks for the author's careful reply. I still think the stochastic ReLU would suffer from a large error. And it is not clear for me to know how you implemented this stochastic ReLU under GC and SS although I carefully read your experimental setting. I found the attached code is only for the python-based unencrypted neural networks instead of core encrypted circuits for your proposed stochastic ReLU. It would be good if the authors could consider releasing the codes regarding the encrypted ReLU.
> > >
> > > For Q2,  if |x| is far smaller than p, it seems that we should use a smaller p to reduce cryptographic overhead in the baseline setting. It would be nice if the authors could explain more the reason why you don't use a smaller p in your experimental baseline setting?  If the baseline can use a smaller p but your proposed method cannot use a smaller p, then it is not a fair comparison.

---

> > > > ### Author Response · Authors · 2021-09-02
> > > > **Clarification on Q1 and Q2**
> > > >
> > > > We thank our reviewer for careful reading of our paper and the opportunity to further clarify our contributions.
> > > >
> > > > **Q1:** The stochastic GC, as we show in our proofs, does incur errors, but with probability $|x|/p$, which is small for the reasons discussed in our previous response and further discussed below.
> > > >
> > > > For a more detailed understanding of the stochastic ReLU operation, the code we attached with the paper includes: (1) a function that functionally emulates the GC protocol used in our stochastic ReLU (`arelu` in `circa_code/Resnet32/resnet.py`); and (2) and our analytical fault model ( `srelu` in `circa_code/Resnet32/resnet.py`). We have compared the `arelu` and `srelu` functions and shown that they produce identical statistical behavior to validate our analytical error model (Fig 3b).
> > > > In the link below, the reviewer will find the actual GC implementation of our stochastic ReLU implemented using `fancy_garbling` library (we should have included this in the code with the paper, that was an oversight on our part and we apologize for that). The `arelu` GC emulator function above is verified to produce the same behavior as the actual GC implementation. \
> > > > https://drive.google.com/file/d/1rre1vxpdRiJyyIz9KlL2LqHvJxbo5fTl/view?usp=sharing
> > > >
> > > > The GC implementation is used to obtain PI runtime numbers, while the `arelu` function (shown to be functionally identical to the GC) is used to obtain accuracy on test data (since running full GC on the test suite would be too slow). We hope that a pointer to the `arelu` and `srelu` functions, along with the underlying GC implementation further clarifies the reviewer’s question.
> > > >
> > > > **Q2:** Prior work (MiniONN, Delphi and others) that we compare with set their $p$ using a *worst-case assumption* that ensures that $p$ is large enough to hold the multiplication of any 15-bit weight 15-bit activation and hence use a 31-bit prime.
> > > > Ours is the first work that looks at the actual *distribution* of inputs ($x$) to the ReLU to reduce GC size, to the best of our knowledge. By focusing on distributions, the reviewer is correct that one can construct an improved baseline that uses a smaller prime that is only as large as the largest $x$ value observed over training/validation inputs. We conducted new experiments to analyze this improved baseline. For VGG16, we observed  that we would need 26-bits and 28-bits to represent the max value for C10/100 and TinyImagenet respectively. At best, this translates to only $1.2\times$ improvement in run-time over Delphi and Circa would still be $2.5\times$ better than this improved baseline. The additional benefits are because of the approximation that Circa introduces in sign computation and via input truncation. We will add these experiments to the paper given the opportunity with due credit to the reviewer’s suggestion. \
> > > > We note that our benefits with respect to other prior work (example SAFENet and DeeReDuce) will be the same as reported in the paper because they all compare with the original Delphi baseline as well.

---

> > > > > ### Comment · Reviewer_DEjr · 2021-09-02
> > > > > **Improved Rating & Question on Stochastic ReLU using fancy_garbling**
> > > > >
> > > > > Thanks for the helpful clarifications. And I improved my rating score from 5 to 7. One question is that I don't find the codes regarding Stochastic ReLU based on fancy_garbling in a short time, although I believe it is included and implemented well. It could be helpful if the authors could add instructions on how to find/run the related codes.

---

> > > > > > ### Author Response · Authors · 2021-09-02
> > > > > > **Thank You**
> > > > > >
> > > > > > Thank you for the constructive discussion and your input that improved our paper! You can find the corresponding GC implementation under `crypto-primitives/src/gc.rs`. We plan on releasing the code publicly along with documentation and instructions.

---

### Official Review · Reviewer_3nef · 2021-07-16

**Rating:** 6
**Confidence:** 1

**Summary:**

The paper proposes several techniques to improve the trade-off between accuracy and storage/runtime for ReLU operations in private inference. The proposed techniques include refactoring ReLUs as a sign function plus a cheap multiplication, and replacing the sign function with a truncated stochastic sign that has runtime and storage beneﬁts (by sacrificing accuracy). Experiments on several datasets and network architectures show that the method has a better trade-off between accuracy and storage/runtime than prior state-of-the-art.

**Limitations And Societal Impact:**

The authors discussed the limitations and potential negative societal impact of their work adequately.

**Main Review:**

I do not work in this area and I am not familiar with related work, so I am not able to judge the originality and significance.

Despite my unfamiliarity with the field, the paper is quite easy to follow. The first two sections provide concise while adequate preliminaries for me to understand the contexts. The approach and evaluation sections are well-organized and the takeaways are clear. Overall, the paper is well-written.

I just have several minor questions/suggestions.
* Line 253-260: It mentions that Circa achieves higher speedup than Delphi (2.6x v.s. 1.2x), but it is unclear what is the accuracy loss of Circa compared with Delphi (1%)?
Line 269: Again, it mentioned that Circa achieves higher speedup than SAFENet (2.6x v.s. 1.9x), but what about the accuracies?
* Line 172: itself.We -> itself. We
* Section 4.2: increasing the spaces between paragraph titles and contents to make it easier to read.

**Time Spent Reviewing:**

3 hours

---

> ### Author Response · Authors · 2021-08-11
> **Response**
>
> We thank our reviewer for their effort and feedback, and we are glad that they found the evaluations well-organized with clear takeaways. We provide an answer to their question in the following.
>
> The baseline accuracy of Delphi/SAFENets ResNet32 model was 0.7% higher than Circa’s (68% vs. 67.3%), which is why we compared relative accuracy drops from baseline using a 1% accuracy drop as a target.
>
> With this in mind, Delphi reports 67.3% accuracy (a 0.7% drop from baseline) with 1.2x speedup, while Circa achieves 66.4% (a 0.9% accuracy drop) with 2.6x speedup. Unfortunately Delphi does not report a datapoint with an exact 0.9% accuracy drop for an ideal comparison, but their next datapoint achieves 66.6% accuracy (a 1.4% drop from baseline) with 1.7x speedup. At this datapoint, Circa has smaller accuracy drop, higher speedup and comparable absolute accuracy vis-a-vis Delphi.
>
> SAFENet reports a 1.9x speedup at 67.5% accuracy (at a 0.5% drop from baseline), while Circa, as above, achieves 2.6x speed-up at 66.4% with a 0.9% drop from baseline. SAFENet unfortunately does not report any datapoints at larger accuracy drops, but we can instead compare SAFENet and Circa at the same speed-up of 1.9x, for which Circa incurs a smaller 0.2% accuracy drop at 67.1% accuracy compared to SAFENets’ 0.5% accuracy drop at 67.5%
>
> Finally, to fairly compare absolute accuracies, we also trained a new ResNet32 model using cosine annealing learning rate for 90 epochs with a 68.15% baseline accuracy that is more competitive with the Delphi/SAFENet baseline. Using this model, Circa achieves 67.76% (a 0.39% accuracy drop) accuracy for 2.6x speedup (17-bit truncation), while SAFENet and Delphi have 67.5% and 67.3% accuracies for 1.9x and 1.2x speedups, respectively. We will add these additional datapoints to the paper.

---

> > ### Comment · Reviewer_3nef · 2021-08-19
> > **Thank you**
> >
> > Thank you for the additional results, which resolved my concerns.

---

### Official Review · Reviewer_auVq · 2021-07-18

**Rating:** 6
**Confidence:** 2

**Summary:**

This paper focuses on minimizing the computation cost of the ReLU operation. Prior work uses Garbled Circuits (GC) to process ReLUs, which is computationally expensive. This paper proposes a novel method to reduce computation by decomposing ReLU operation into a sign computation and multiplication. The multiplication can be implemented by Secret Sharing and the cost of sign computation through GC can be reduced by approximation. This paper also provides adequate empirical evaluations to show the proposed methods can reduce runtime.


**Limitations And Societal Impact:**

Yes, they discuss them in Section 6.

**Main Review:**

The idea of decomposing ReLU into multiplication and sign computation is nice. I also like the idea of approximating the sign computation to reduce the computation cost. The authors also provide the theoretical analysis of the fault rates of the stochastic approximation. The experiment evaluation is adequate. The authors evaluate Circa on various deep learning models and the results show the Circa requires much less runtime. This paper is also well-written and organized.

Minor comments and questions:
1.	In the experiments, could the authors elaborate on how those models are trained? Does training those models involves Circa or it is used only in evaluating the test accuracy?
2.	Line 163: Typo in ‘This multiplication is cheap and the ReLU optimized ReLU’.



**Time Spent Reviewing:**

8

---

> ### Author Response · Authors · 2021-08-11
> **Response**
>
> We thank our reviewer for their effort to review our manuscript, and we are glad they like the main ideas and accompanying theoretical analysis. We address their question below.
>
> All models were trained normally in plaintext (Circa is not involved). ResNet32 models on C10 were obtained from this [repository](https://github.com/akamaster/pytorch_resnet_cifar10). We trained the rest of the networks from scratch with the same training procedure used for ResNet32 in the aforementioned repo (more precisely,  SGD with step learning optimizer with 0.1 initial learning rate, 128 batch size, 0.0001 weight deacy, 0.9 momentum and milestones at 100th and 150th epochs for 200 epochs). The models are trained in floating point and the model parameters are then quantized and converted to 15-bit fixed point values. Circa was only used in inference, and interestingly performs well even over already available and commonly trained networks. We will further clarify this in the paper.

---

> > ### Author Response · Authors · 2021-08-31
> > **Available for Discussion**
> >
> > We hope our responses have addressed the reviewers concerns, but if not we are available/open to addressing any outstanding issues.

---

### Decision · Program_Chairs · 2021-09-27

**Decision:**

Accept (Poster)

**Comment:**

Thank you for your submission. The reviewers agree that this paper provides new techniques to reduce the computational overhead of ReLU-based private inference. During the discussion, the authors have addressed the questions raised by the reviewers. The authors should incorporate these clarifications during rebuttal into the next revision of the paper.